# The Mediating Role of Academic Self-Efficacy Between Gender and the Intention to Resume Schooling Among Depressed Adolescents: An Empirical Study from China

**DOI:** 10.3390/bs15070963

**Published:** 2025-07-16

**Authors:** Ruiling Zhang, Chun Liu, Min Yang

**Affiliations:** Xiangya School of Nursing, Central South University, Changsha 410013, China; 237811039@csu.edu.cn (R.Z.); 217811043@csu.edu.cn (C.L.)

**Keywords:** resume schooling, academic self-efficacy, suspension of schooling, depression, adolescent, intention

## Abstract

Depressed adolescents often face challenges in resuming schooling, with gender differences and psychological factors influencing their intentions. This study aimed to investigate the mediating role of academic self-efficacy in the relationship between gender and the intention to resume schooling. Using convenience sampling, 295 school-suspended depressed adolescents (aged 12–18; mean age 15.84 ± 1.49; 75 males, 220 females) completed the General Situation Questionnaire, Questionnaire on the Intention to Resume Schooling, Kutcher Adolescent Depression Scale, and Academic Self-efficacy Questionnaire. Data were analyzed via SPSS 27.0.1 and R 4.4.3 software. Mediation analysis was performed using logistic regression modeling. The results showed that 42.70% (126/295) of participants intended to resume schooling. Gender disparities emerged in both intention to resume schooling (χ^2^ = 18.63, *p* < 0.001) and academic self-efficacy (*Z* = 2.31, *p* < 0.001) among depressed adolescents. Additionally, academic self-efficacy significantly differed across intention to resume schooling (*Z* = 3.05, *p* < 0.001). Gender differences in resumption intention were partially mediated by academic self-efficacy (effect = −0.18, 42.86% of total effect), with a direct gender effect of −0.24 (57.14%). These findings suggest that the gender of depressed adolescents not only directly affects their intention to resume schooling, but also indirectly affects their intention to resume schooling through academic self-efficacy.

## 1. Introduction

Depression is a common psychological disorder characterized by high prevalence and disability ([43]). Adolescent depression is becoming increasingly prevalent ([18]). A 2022 meta-analysis of 96 studies across 29 countries (N = 528,293; 1989–2022) estimated pooled prevalence rates of depressive disorders in children and adolescents as 21.3% (95%CI: 16.7–26.7%) for mild-to-severe cases, 18.9% (95%CI: 14.6–24.2%) for moderate-to-severe cases, and 3.7% (95%CI: 2.7–5.1%) for major depressive disorder ([29]). Symptoms of depression, including social impairment, loss of motivation, disrupted sleep patterns, and diminished energy, negatively impact adolescents’ ability and intention to attend school, leading to a widespread occurrence of school suspension. Studies indicate ([58]) that psychological disorders account for 26.58% of all school suspensions, with depression being the most common, comprising 84.52% of suspensions due to psychological disorders. Research also shows that suspension from school may further undermine the already fragile school relationships of depressed adolescents, who are more likely to feel marginalized and discriminated against, as well as experiencing a decline in academic performance ([42]). In addition, school suspension has been linked to risky behaviors, including tobacco use, criminal activity, and violent behavior ([32]). In order to reduce the negative consequences of school suspension for depressed adolescents, it is imperative to promote the resumption of schooling for depressed adolescents. Studies have shown that patients with depression can achieve functional recovery through individualized, comprehensive, and recovery-oriented interventions, offering the possibility of returning to school ([47]). Moreover, the resumption of schooling can also promote recovery from depression and improve patients’ social functioning and quality of life.

Jiang Shining ([17]) posits that the process of resuming schooling is a cohesive, systematic, and dynamic sequence that can be delineated into three distinct stages based on the timeline and progression of development. Initially, the stage following the adolescent’s refusal to attend school is termed the acute treatment stage; subsequently, the stage of preparing to resume schooling is denoted as the pre-return stage; and ultimately, the stage of gradual acclimation and stabilization post-resume schooling. For adolescents with depressive disorders, the stage following school refusal is predominantly characterized by the acute treatment phase, where the focus is on therapeutic intervention rather than academic pursuits. As treatment progresses, the patient’s condition stabilizes, and the restoration of social functioning, including resuming schooling, gains prominence, signifying the transition to the preparation stage for school re-entry. The stage of gradual acclimation and stabilization subsequent to returning to school is marked by ongoing adjustments in response to continuous treatment, emphasizing the integration and adaptation to school life while maintaining continuous treatment. Given the distinct foci of each stage, it is challenging to encompass all stages within a single study. Consequently, this investigation concentrates on the second stage, the pre-return stage, during which the patient’s condition and social functioning progressively improve.

Previous research on school resumption has focused on populations who suspend school due to somatic illnesses, such as cancer, brain injury and chronic illnesses ([1]; [11]; [35]). However, only a few scholars have focused on the resumption of schooling for depressed adolescents, and their studies have focused on the development of transition programs for resumption ([48]; [53]) and the identification of eligibility for resumption after a break from school ([49]). Most of these transition programs have emphasized only external factors, such as school, family, and social support, but there has been limited exploration of the internal factors of depressed adolescents. Only very few researchers have used qualitative research methods to explore resumption experiences and needs ([38]; [39]), but no studies have focused on the intentions of depressed adolescents to resume schooling. Most studies on the resumption of schooling after a break are reviews or response analyses, and empirical studies are lacking. Therefore, this study conducted empirical research on the intention of depressed adolescents to resume schooling.

Intention is conceptualized as the psychological state in which an individual desires to engage in a behavior prior to its execution, and it exerts influence over an individual’s decision-making and the probability of behavior enactment ([55]). This concept encompasses both immediate and prospective intentions ([34]). Immediate intentions pertain to the momentary impulses or desires that an individual experiences, which may be transient or impulsive. In contrast, prospective intentions involve the long-term goals and aspirations that individuals formulate, typically linked to intentional decision-making and strategic planning. In conjunction with the above conceptualization of intention, the intention to resume schooling in this study also includes immediate and prospective intentions. Immediate intention is assessed by inquiring whether participants are willing to resume schooling. Prospective intention is evaluated by examining whether participants have a plan for resuming schooling and have initiated preparations for doing so. The intention to resume schooling is operationally defined as encompassing the willingness to return, the presence of a plan for returning, and the initiation of preparations for school resumption.

Epidemiological characteristics of depressed adolescents indicate a higher prevalence rate among females compared to males ([2]). Previous studies have predominantly examined male and female students as a homogeneous group, with limited attention to gender-specific manifestations. However, existing research demonstrates significant gender differences in students’ learning motivation ([40]). Nayak’s findings establish that individual motivation significantly influences behavioral intentions ([33]), suggesting potential gender-based variations in intention to resume schooling. Furthermore, depressive disorders typically present with the classic triad of symptoms (low mood, psychomotor retardation, and anhedonia) ([3]), which invariably diminish volitional capacity and consequently reduce intention to resume schooling. This raises a critical research question: Do gender differences in school re-entry intentions persist among clinically depressed adolescents after controlling for symptom severity? In recent years, educational researchers have paid increasing attention to the role of students’ thoughts and beliefs in the learning process ([54]). Self-efficacy is one of the most important personal beliefs. The manifestation of self-efficacy within the academic domain is called academic self-efficacy. Academic self-efficacy refers to an individual’s beliefs about learning ability, i.e., an assessment of how confident an individual is in his or her ability to use his or her skills to accomplish learning goals and is a subjective judgment ([4]). Academic self-efficacy includes self-efficacy of learning ability and self-efficacy of learning behavior. Self-efficacy of learning ability refers to an individual’s belief in their cognitive capacity to successfully complete domain-specific academic tasks (e.g., “Can I solve this math problem?” or “Am I capable of writing a high-quality essay?”). Self-efficacy of learning behavior reflects an individual’s confidence in their ability to regulate and sustain learning-related actions (e.g., “Can I maintain a consistent study schedule?” or “Will I resist distractions while preparing for exams?”). According to self-efficacy theory ([62]), a person’s self-efficacy directly affects behavioral intentions. Therefore, for depressed adolescents who have taken a break from school, their academic self-efficacy will have a direct impact on the intention to resume schooling. Furthermore, research results indicate significant gender disparities in academic self-efficacy among college students ([45]), and Qiao’s research also demonstrates that self-efficacy typically functions as a mediating factor ([41]). Therefore, this study aims to investigate the mediating role of academic self-efficacy in the relationship between gender and the intention to resume schooling among depressed adolescents.

In summary, the research hypotheses of this study are as follows: (1) there are gender differences in the intention of depressed adolescents to resume schooling; (2) depressed adolescents exhibit gender differences in academic self-efficacy; (3) depressed adolescents show differences in academic self-efficacy based on their intention to resume schooling. (4) academic self-efficacy is a mediating variable between gender and intention to resume schooling.

## 2. Methods

### 2.1. Design and Participants

This cross-sectional study used convenience sampling to select samples. Participants were recruited from eligible recovering depressed adolescents in two tertiary hospital mental health centers in Changsha, Hunan Province, China, during the period from March 2023 to December 2023. Inclusion criteria: (1) patients diagnosed with depressive disorder according to the International Classification of Diseases of the World Health Organization (ICD) or depression according to the Chinese Classification and Diagnostic Criteria for Mental Diseases-III (CCMD-III); (2) aged 12–18 years old; (3) during school suspension period (if the patients did not apply for the off-semester period, consecutive sick leave of more than 3 months can be regarded as the off-semester period); (4) parents’ informed consent was obtained, and the study participants volunteered to take part in the study. Exclusion criteria: (1) patients who had a seizure during the study period; (2) patients with a disease duration of less than 3 months; (3) patients with major depression and psychotic symptoms; (4) patients with other organic diseases; (5) patients who refused to cooperate after the researcher’s explanation; (6) patients with comorbid psychiatric disorders (e.g., personality disorders).

Sample size was calculated using G*Power 3.1.9.7 ([9]). The sample size, with 80% power, a 0.05 significance level (α), a medium effect size of 0.16, and 2 predictor variables, was estimated to be 64 participants. In addition, considering a 20% sample attrition rate, the final minimum sample size was 80. Initially, 348 individuals participated in this study. After exclusion, 295 valid participants remained, ensuring an adequate sample size. We excluded 17 respondents who answered questionnaires randomly and 36 with major depression, as indicated by their depression scores. The demographic characteristics in this study are shown in Table 1. The 295 participants had a mean age of 15.58 ± 1.49 years, with 75 males and 220 females. Median depression scores were 17.83 (IQR, 14.00–23.00) for males and 19.40 (IQR, 14.00–23.00) for females. A Kolmogorov–Smirnov test found no significant gender difference in depression scores (*Z* = 1.126, *p* = 0.158).

### 2.2. Data Collection

Data were collected using a structured questionnaire that was administered by a doctor with experience in research related to adolescent depression. Prior to commencing the study, all participants were educated about the objectives and procedures of the research, and they were assured of the anonymous reporting of the findings. Participants provided written informed consent before proceeding to complete a set of questionnaires. During the questionnaire completion process, the researcher was present in the same room as the participant, offering clarifications and support as needed. The questionnaires were distributed on the spot and returned on the spot, and then the integrity of the questionnaire information was verified on the same day by the research team, and the data were entered into IBM SPSS statistics version 27.0.1.

### 2.3. Measures

#### 2.3.1. The General Situation Questionnaire

The questionnaire included sociodemographic characteristics related to intention to resume schooling. It was developed through a review of the literature and discussions in our research group. This questionnaire specifically included entries such as gender, age, place of residence, whether or not the child was an only child, current grade level attended, and per capita monthly household income.

#### 2.3.2. The Questionnaire on the Intention to Resume Schooling

The questionnaire on the intention to resume schooling was used to assess the patient’s intention to resume schooling (see Appendix A). The questionnaire was developed by the research team based on a review of the literature ([34]; [56]). The questionnaire included three questions: whether they were willing to resume schooling on schedule, whether they had made plans to resume schooling (e.g., planned the time of resumption, resumption school, etc.), and whether they had prepared for resuming schooling (e.g., adjusted their work schedules, pre-studied their homework, and prepared themselves psychologically, etc.). The Cronbach’s alpha coefficient of the questionnaire was 0.702, and the coefficient results were acceptable. In this study, patients who were “willing to resume schooling on schedule” and “made a plan for resumption of schooling” and “have begun to prepare for resumption of schooling” were considered to “have an intention to resume schooling” and the others were considered to “lack an intention to resume schooling”.

#### 2.3.3. The Kutcher Adolescent Depression Scale (KADS-11)

The depression level of the patients was evaluated using the KADS-11. We used the validated Chinese version of KADS-11 ([61]), which consists of 11 entries, each reflecting a core symptom of depressive mood, such as depressed mood, impatience, and sleep difficulties. Each entry was scored on a 4-point Likert scale, ranging from “almost never” to “all the time”, with scores ranging from 0 to 3, respectively. If the total score is ≥9, it indicates that there are depressive symptoms, and the higher the total score, the more serious the depressive symptoms. The reliability of the scale was good, with a Cronbach’s alpha coefficient of 0.84. The norm score of the KADS-11 for Chinese depressed adolescents is (15.5 ± 5.9) ([61]). According to the mean standard deviation method ([30]), depressed adolescents with scores higher than 27.3 (15.5 + 5.9 × 2) were classified as having major depressive disorder, and those with major depressive disorder were excluded.

#### 2.3.4. The Academic Self-Efficacy Questionnaire

The academic self-efficacy questionnaire was used to evaluate the academic self-efficacy of the patients. The questionnaire was developed by Pintrich and De Groot ([36]), which consists of 22 items, including two dimensions: learning ability and learning behavior. Each item was rated on a 5-point Likert scale, with scores ranging from 1 to 5, from “not at all consistent” to “completely consistent”. The reliability of the Chinese version of the questionnaire was good, and the Cronbach’s alpha coefficient was 0.88 ([27]).

### 2.4. Statistical Analyses

SPSS 27.0.1 software was used for data processing and analysis. The Kolmogorov–Smirnov test was used to examine the normality of the quantitative data. The results of the test showed that the data did not conform to a normal distribution (*p* < 0.05), so quantitative data with a non-normal distribution were described by M (IQR, P25-P75). Two-sample Kolmogorov–Smirnov and chi-square tests were used for the comparison between groups. R 4.4.3 software was used to establish and test the mediation model. Firstly, the distribution-of-product method was used to determine whether the mediation effect was established or not, and then the Mediation package was used to calculate the mediation effect value (in which the intention to resume schooling and gender were dichotomous variables, so the logistic regression model was used for the mediation analysis in the analyzing process). To address the issue of sample imbalance, we applied inverse probability weighting to assign differential weights by gender in the mediation analysis model, thereby adjusting for the influence of gender distribution. Finally, the stability of the mediation model was tested using the sensitivity analysis. The dichotomous variables were assigned values of male = 0, female = 1, no intention to resume schooling = 0, and intention to resume schooling = 1. The significance level (α) of the test was set at 0.05.

## 3. Results

### 3.1. Current Status of Intention to Resume Schooling

According to the concept of intention to resume schooling, participants who had both the willingness to resume schooling, had a plan to resume schooling and made preparations for resuming schooling were defined as the group with intention to resume schooling, and there were 126 cases in this study, accounting for 42.70%. The details of whether they were willing to resume schooling, had a plan to resume schooling and made preparations for resuming schooling are shown in Table 2.

### 3.2. Analysis of Differences in Intention to Resume Schooling by Gender

Among participants, 42.71% expressed intention to resume schooling. Analysis by gender revealed that 64.00% of male participants endorsed this intention, compared to 35.45% of female participants. A chi-square test indicated that this gender difference was statistically significant, χ^2^ = 18.626, *p* < 0.0001, with males showing greater school return intention than females (see Table 3).

### 3.3. Analysis of Differences in Academic Self-Efficacy by Gender

There was a statistical difference in the academic self-efficacy scores of depressed adolescents by gender, with males having higher academic self-efficacy scores than females, as shown in Table 4.

### 3.4. Analysis of Differences in Academic Self-Efficacy by Intention to Resume Schooling

The academic self-efficacy of depressed adolescents with and without intention to resume schooling was different, and the difference was statistically significant, i.e., the academic self-efficacy of depressed adolescents with intention to resume schooling was stronger. In other words, the intention to resume schooling increases with the increase in academic self-efficacy (see Table 5).

### 3.5. Intermediary Analysis

#### 3.5.1. Mediation Effects

The mediation analysis in this study utilized the distribution-of-product method, which does not require normality assumptions, is suitable for small to medium samples, and can be run automatically using the RMediation package of R 4.4.3 software ([52]). The results showed a 95% confidence interval for the mediating effect of academic self-efficacy between gender and intention to resume schooling is (−0.663, −0.216), without the confidence interval including 0, indicating that the mediation effect of academic self-efficacy between gender and intention to resume schooling is significant. The Mediation package of R 4.4.3 software was used to calculate the mediated effect value, while the Survey package was used to implement the inverse probability weighting method. The calculation yielded an indirect effect value of −0.18, with a 42.86% share of indirect effects; a direct effect value of −0.24, with a 57.14% share of direct effects; and a total effect value of −0.42 (Table 6). The mediation model of this study is shown in Figure 1.

#### 3.5.2. Model Test

The logistic regression analysis employed in this study achieved a convergence rate ranging from 96.28% to 100%. The model converged under the majority of conditions; however, convergence was not achieved under specific circumstances, such as when the sample size was minimal (<50) or when the mediation effect was pronounced (e.g., standardized coefficients of b and c’ were 0.59). None of these specific conditions were present in the current study, hence the model converged successfully.

Figure 2 shows a sensitivity analysis plot in which the contour lines indicate the true ACME (indirect effect), plotted as a function of the ratio of the total mediating variable variance (horizontal axis) and the total outcome variance (vertical axis), which are both explained by unobserved confounders included in the corresponding regression model. Here it is assumed that unobserved confounders affect mediators and outcomes in the same direction. The plot was generated by the R 4.4.3 software, and the thick line in the figure indicates that when the product of the raw variances explained by the omitted confounders is 0.06, the point estimate of the ACME will be 0. The figure shows that when the product of these proportions is greater than 0.06, the true ACME changes sign, from a negative sign to a positive sign, and that the confounders affect gender and intention to resume schooling in the opposite direction. Because unmeasured confounders explain less than 24% (√0.06) of the variance in the mediation and outcomes ([16]), the ACME reported in the original analysis is robust to unmeasured confounders.

## 4. Discussion

### 4.1. Gender Ratio and Intention to Resume Schooling Rate

This study reveals a significantly higher prevalence of female students among suspended depressed adolescents compared to male students. This finding echoes the findings of existing studies, which generally state that females are approximately two times more likely to be depressed than males ([60]). First, in terms of cognition, females tend to ruminate more than males when faced with stressful events ([19]). This causes females to focus their attention on and ruminate over negative stressful events rather than taking action to alleviate the distress. Additionally, in terms of physiological factors, changes in reproductive hormones specific to females have all been associated with the development of depression, and changes in physiological hormones may lead to mood swings, which can increase the prevalence of depression in females ([21]). This finding further emphasizes the need to pay special attention to the mental health status of female adolescents in the field of adolescent mental health.

This study found that less than half of depressed teens who were on a break from school wanted to resume schooling. No study has yet examined the intention of depressed adolescents to resume schooling or the rate of resuming schooling, but the intention of depressed adolescents to resume schooling appears low when compared to the rates of resuming schooling for children who survived tumors (56%) ([10]) and for children who refused to go to school (67%) ([37]). Compared to tumor survivors and school refusers, the challenges faced by depressed adolescents are mainly mental health issues, which will directly affect their academic motivation and social skills. In contrast, the challenges faced by children who are tumor survivors are mainly physical health issues ([22]), and their process of resuming school tends to focus more on physical rehabilitation and adjustment to normal school life ([31]). Social support and understanding for children who survived tumors is also more explicit, and their teachers and classmates tend to provide more care and assistance when they resume schooling ([51]). The school refusal behavior of school refusal children is caused by a variety of factors ([6]), including both internal factors such as depression, timidity, aggressiveness, behavioral withdrawal, etc., and external factors such as excessively strict parental requirements, family conflicts, high study pressure, deterioration of relationships with teachers and classmates, Internet addiction, and so on. When the external factors of school refusal are eliminated, children with external causes demonstrate an intent to resume schooling ([28]). However, internal factors are not easy to address. Pleasure loss is one of the core symptoms of depressed adolescents. Moreover, studies have shown that depressed adolescents with pleasure loss have an obvious lack of motivation, and some depressed adolescents even show comprehensive motivational inhibition and have no interest in anything ([50]), so depressed adolescents’ intrinsic motivation to learn is weak ([15]), and their intention to return to schooling is also weak. This suggests that besides improving external influencing factors, it may be a greater challenge for researchers to enhance the internal motivation of depressed adolescents.

### 4.2. Gender Differences in the Intention to Resume Schooling

The findings indicate that male adolescents with depression who have been suspended exhibit a greater intention to resume schooling than their female counterparts. Adolescents with depression who have been suspended may confront heightened academic stress due to an accumulation of missed lessons, and the struggle to align with the curriculum can intensify their depressive symptoms ([8]). Some studies have found that in terms of learning pressure, females with depression find it more difficult to cope with various pressures after taking a leave of absence from school ([12]), and as a result, they may demonstrate lower school resumption intentions. Moreover, the influence of socialization should not be dismissed. Adolescent females tend to be more emotionally sensitive than males, place higher value on others’ perceptions, and exhibit a stronger need for social interaction ([46]). Upon re-entering the school environment following a suspension, adolescents with depression often encounter significant challenges in social interactions ([5]), such as concerns over others’ perceptions, uncertainty about how to account for their absence, or potential victimization due to their mental health condition or hospital stay. The aforementioned factors may collectively contribute to the diminished intention among depressed female adolescents to resume schooling compared to their male peers. This study suggests that schools are recommended to implement comprehensive mental health support programs in educational policies. For students who are suspended due to depression, schools should develop individualized support plans that take into account their gender-specific needs. For example, for female students, the program could focus more on building emotional resilience and providing a safe space to express concerns about resuming school. In addition, given the academic pressures faced by adolescents who resume school due to depression, schools are encouraged to adapt the curriculum for these students. This may include providing additional tutoring, allowing more flexible schedules to make up for missed classes, and modifying assignments to make them easier to complete.

### 4.3. Gender Differences in Academic Self-Efficacy

The investigation revealed that male students exhibited higher academic self-efficacy than their female counterparts among suspended depressed adolescents, specifically in both academic ability and academic behavior self-efficacy. This is consistent with the findings of Li and Huang ([14]; [25]). Male secondary school students had stronger academic self-efficacy total scores and sub-dimensions than female students. However, in terms of academic self-efficacy scores, suspended depressed adolescents were lower than regular secondary school students, i.e., the academic self-efficacy of suspended depressed adolescents was lower than that of regular adolescents, which is consistent with Justin’s study ([20]). Bandura’s theory of self-efficacy ([4]) posits that emotional state is a critical determinant of academic self-efficacy. Regarding emotional regulation, males are found to be more proficient in managing negative emotions than females ([44]), and females are more vulnerable to negative events, leading to a greater likelihood of diminished academic self-efficacy when encountering academic pressures. This suggests that researchers can target the level of emotion regulation to improve the academic self-efficacy of depressed female adolescents.

### 4.4. Differences in Academic Self-Efficacy Based on Their Intention to Resume Schooling

This study reveals that the intention to resume schooling among depressed adolescents who took a break from school is positively correlated with their academic self-efficacy. This result is consistent with the findings of Preyde ([39]), who noted that people with mental illness are highly concerned about academic competence before resuming school, which will directly influence their intention to resume their education. An empirical study showed that adolescents’ self-efficacy was associated with intentions to achieve good grades ([13]). Moreover, self-efficacy theory, which also suggests that self-efficacy will directly influence behavioral intentions, corroborates this conclusion. The findings suggest that enhancing the academic self-efficacy of depressed adolescents can serve to bolster their intention to resume schooling, consequently improving the likelihood of their re-enrollment. For teachers, it is possible to use encouraging teaching and provide positive feedback to enhance students’ academic self-efficacy.

### 4.5. Mediating Role of Academic Self-Efficacy

Academic self-efficacy among depressed adolescents who had taken a break from school was found to partially mediate the relationship between gender and the intention to resume schooling. This indicates that, for depressed adolescents who had taken a break from school, gender influenced the intention to resume schooling via academic self-efficacy. The findings revealed that male depressed adolescents exhibited higher academic self-efficacy, which correspondingly led to a stronger intention to resume schooling. Influenced by traditional Chinese gender role norms, academic performance in secondary school is commonly viewed as a significant predictor of future career success and economic proficiency ([24]). Consequently, academic performance during this period emerges as a critical factor in assessing the value of an individual’s gender role ([26]). It is generally expected by society that males will exceed females in terms of economic proficiency, leading to encouragement for males to demonstrate greater confidence and competence academically during adolescence ([59]). As a result, in the depressed adolescent group, males have greater academic self-efficacy and thus greater intention to resume schooling in order to achieve better academic results, thus conforming to societal expectations of economic competence in relation to their gender roles ([23]). As previously discussed, the academic self-efficacy of adolescent females is diminished due to factors such as sensitivity, skepticism, and a lower resilience to stress, and their intention to resume schooling is similarly influenced by their level of self-efficacy. This suggests that educators and mental health experts should not only pay attention to the influence of gender differences on the intention of depressed adolescents to resume schooling, but also adopt different methods for different genders while improving self-efficacy. Given that prior research suggests females may rely more heavily on social sources of self-efficacy ([45]), a promising approach to enhance female students’ academic self-efficacy involves teachers providing more positive encouragement and feedback ([57]). Educators should avoid gender-stereotyped evaluations (e.g., ‘Girls are naturally worse at science’) and instead adopt growth-mindset language (e.g., ‘Math abilities can improve with practice’) ([7]).

## 5. Limitations and Prospects

There are five limitations of this study. First, the findings cannot be generalized to all depressed adolescents because the sample only included depressed adolescents in Hunan, China, and academic self-efficacy is affected by cultural differences, especially the large differences in socio-cultural background between domestic and foreign countries, and the applicability of the findings to depressed adolescents in foreign countries needs to be verified by other studies. Second, there are many influences on the intention to resume schooling, and only two of these influences were explored in this study; future studies should investigate more influences and the interactions between factors. Third, the number of females in this study of depressed adolescents was more than three times that of males, which may be related to the small sample size and single sampling location in this study, and it is recommended that future studies sample at different locations and increase the sample size. Fourth, the cross-sectional nature precludes causal conclusions. We recommend longitudinal studies in future research to clarify temporal dynamics. Fifth, while our sensitivity analysis indicated relatively small confounding effects, a key limitation remains our inability to account for the etiological heterogeneity of depression. The distinction between school-related (e.g., academic stress, bullying) and external (e.g., familial issues, personal trauma) depression sources may differentially influence students’ motivation to resume schooling, with some perceiving school as traumatic and others as supportive. Future studies should incorporate detailed assessments of depression sources to better disentangle these potential mechanisms.

## 6. Conclusions

This study explored the relationship between gender, academic self-efficacy, and intention to resume schooling among depressed adolescents, demonstrating the following: first, the intention to resume schooling and academic self-efficacy of depressed adolescents are low and need to be improved; second, for depressed adolescents who had taken a break from school, the intention to resume schooling and academic self-efficacy are stronger for males than for females, and the intention of depressed adolescents who have a high level of academic self-efficacy to resume schooling is even higher; and third, academic self-efficacy partially mediates the relationship between gender and intention to resume schooling.

## Figures and Tables

**Figure 1 behavsci-15-00963-f001:**
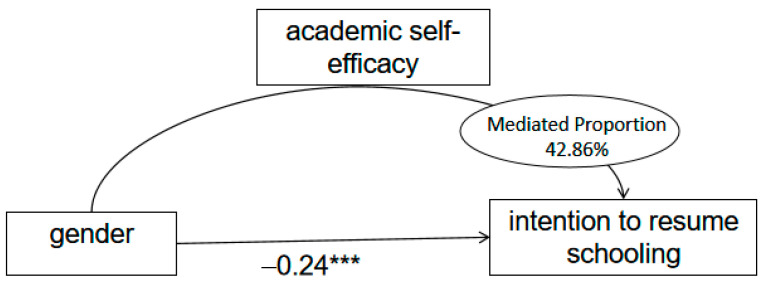
Mediation effect model (N = 295). *** *p* < 0.001.

**Figure 2 behavsci-15-00963-f002:**
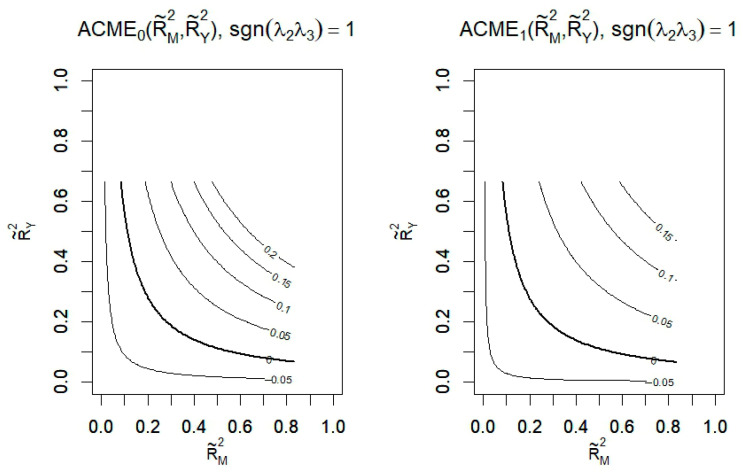
Sensitivity analysis (N = 295).

**Table 1 behavsci-15-00963-t001:** Demographic characteristics of participants (N = 295).

Variables						
Age (M, SD)	15.58 (1.49)					
Gender	Male (25.42%)	Female (74.58%)				
Place of residence	Urban (72.54%)	Rural (27.46%)				
Only one child	Yes (39.66%)	No (60.34%)				
Monthly household income per capita (CNY)	<2000 4.75%	2000~4000 28.81%	4000~6000 33.22%	>6000 33.22%		
Grade	Grade 7 7.46%	Grade 8 13.56%	Grade 9 15.59%	Grade 10 26.44%	Grade 11 26.78%	Grade 12 10.17%

**Table 2 behavsci-15-00963-t002:** Classification of intention to resume schooling (N = 295).

Variables	N	%
**Willingness**		
No	100	33.90
Yes	195	66.10
**Plan**		
No	103	34.90
Yes	192	65.10
**Preparation**		
No	118	40.00
Yes	177	60.00
**Intention to resume school**		
No	169	57.30
Yes	126	42.70

**Table 3 behavsci-15-00963-t003:** Gender differences in the intention to resume schooling (N = 295).

		Gender		Total
		Male	Female	
**Intention to resume school**	No	27	142	169
	Yes	48	78	126
**Total**		75	220	295
**Intention to resume rate (%)**		64.00%	35.45%	42.71%
**χ^2^ value**		—	—	18.626
***p* value**		—	—	<0.001

**Table 4 behavsci-15-00963-t004:** Gender differences in academic self-efficacy (N = 295).

	Gender	N	M	(P25–P75)	*Z*-Value
**Academic self-efficacy**	Male	75	66.00	(52.00–75.00)	2.314 ***
	Female	220	53.00	(43.00–63.75)	
**Self-efficacy of learning ability**	Male	75	33.00	(24.00–41.00)	2.452 ***
	Female	220	23.00	(17.25–31.00)	
**Self-efficacy of learning behavior**	Male	75	32.00	(28.00–38.00)	0.019 *
	Female	220	29.00	(24.25–34.00)	

Note. * *p* < 0.05, *** *p* < 0.001.

**Table 5 behavsci-15-00963-t005:** Differences in academic self-efficacy based on their intention to resume schooling (N = 295).

	Intention to Resume School	N	M	(P25–P75)	*Z*-Value
**Academic self-efficacy**	No	169	49.00	(38.00–62.00)	3.053 ***
	Yes	126	63.00	(52.75–75.00)	
**Self-efficacy of learning ability**	No	169	22.00	(16.00–29.50)	2.713 ***
	Yes	126	31.00	(22.00–37.25)	
**Self-efficacy of learning behavior**	No	169	28.00	(23.00–33.00)	2.703 ***
	Yes	126	33.00	(29.00–38.25)	

Note. *** *p* < 0.001.

**Table 6 behavsci-15-00963-t006:** Mediating effect of academic self-efficacy between gender and intention to resume schooling (N = 295).

	Path	Efficiency Value	95%CI	Proportion of Total Effect (%)
**Indirect effect**	Gender → Academic self-efficacy → Intention	−0.18 ***	−0.22~−0.07	42.86%
**Direct effect**	Gender → Intention	−0.24 ***	−0.37~−0.11	57.14%
**Total effect**	Gender → Intention	−0.42 ***	−0.51~−0.24	100.00%

Note. *** *p* < 0.001.

## Data Availability

The data presented in this study are available on request from the corresponding author. The data are not publicly available due to restrictions, e.g., privacy or ethical concerns.

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
