# Peer review of "The Mediating Role of Academic Self-Efficacy Between Gender and the Intention to Resume Schooling Among Depressed Adolescents: An Empirical Study from China"

_behavsci, 2025, doi:10.3390/bs15070963_

Round 1

Reviewer 1 Report

Comments and Suggestions for Authors

It was a very exciting topic.

The abstract contains all the important information.

The theoretical background is based on a lot of literature and research, but it should be much better synthesised – showing the main results of the research and the differences between them (as many tools, so many results). Also, much more recent research should be included in the theoretical background.

There are some typos in the text (e.g., space before brackets is missing in several places). The notation of statistical data is not precise (e.g., not P). There should be no double spaces in the interlinear references (e.g. line 206, Pintrich...). The figures are very small, cannot be seen, this should definitely be changed. They are useless in this form. 

The description of the IRS is not precise, it should be detailed (e.g., development process, results, detailed psychometric indicators). 

No justification is given as to why these data are included in the GSQ. This is needed, as they are used to interpret the results obtained in the questionnaires (or should be along these lines in much more detail).

In the summary, there are a lot of statements that are not linked to a reference. It should be clearly stated whether or not a result is related to previous data. If the own data is analysed and not linked to a previous result, this should be made explicit.

They are dealing with a very important topic, but this study is not yet ready. I hope I have been able to help with these points to produce even better work.

Comments on the Quality of English Language

The English could be improved to more clearly express the research.

Author Response

Thank you very much for your time and valuable comments on our manuscript. We sincerely appreciate your insightful suggestions, which have significantly improved the quality of our work. We have carefully addressed all of your concerns in the revised manuscript. All modifications have been highlighted in red text for your convenience. Additionally, please find attached our point-by-point response document detailing how we have implemented each of your suggestions.

Reviewer 2 Report

Comments and Suggestions for Authors

This study addresses a valuable research topic with practical implications for educational settings; however, several areas require improvement. 
First, while the abstract focuses on simple research findings and technical descriptions, this paper actually deals with a multilayered influence structure rather than simple causality. Gender should not be presented merely as a demographic variable, but rather as representing differences arising from complex factors associated with gender disparities.
Second, both the abstract and main text need methodological descriptions that go beyond simply listing software used. The rationale for why such approaches are necessary should be articulated in connection with the research questions, and unnecessary content in the abstract should be removed.
Third, the literature review should be grounded in theoretical frameworks and connected to both the research questions and results. This would strengthen the overall coherence of the study.
Fourth, various dimensions of self-efficacy are mentioned, and it is necessary to emphasize their differences by supplementing conceptual and operational definitions to clarify these distinctions.
Fifth, the research questions, results, and discussion should be restructured to enhance connectivity and flow throughout the paper, ensuring all sections are logically linked.
Sixth, errors in result interpretation and the accuracy of methodological descriptions require thorough re-examination and correction. 
Seventh, adding educational intervention strategies to the discussion of these research findings would provide practical benefits for educational researchers. For example, how should academic re-enrollment support strategies differ by gender? What answers can this study provide to this question? What are the anticipated effects of such approaches?
The finding that students with higher self-efficacy have greater intention to resume their studies may be a rather predictable result. This research would become more compelling if it could identify and discuss aspects that people might not intuitively consider, through more nuanced analysis and results that go beyond common expectations.

Author Response

(The authors gave the same response as above.)

Round 2

Reviewer 1 Report

Comments and Suggestions for Authors

I am satisfied with the improvements. The language of the study could be improved.

Comments on the Quality of English Language

The English could be improved to more clearly express the research.

Author Response

Thank you for your valuable feedback and constructive suggestions regarding my manuscript. I sincerely appreciate the time and effort you have taken to review my work and provide guidance for improvement.

In response to your comments, I have carefully revised the English language usage. To ensure clarity while preserving the original meaning, I sought assistance from a professional language editor. I hope the revised version meets the journal’s standards and addresses your concerns.